# Role of Dual-Staining p16/Ki-67 in the Management of Patients under 30 Years with ASC-US/L-SIL

**DOI:** 10.3390/diagnostics12020403

**Published:** 2022-02-04

**Authors:** Cristina Secosan, Andrea Pasquini, Delia Zahoi, Andrei Motoc, Diana Lungeanu, Oana Balint, Aurora Ilian, Ligia Balulescu, Dorin Grigoras, Laurentiu Pirtea

**Affiliations:** 1Department of Obstetrics and Gynecology, Victor Babeş University of Medicine and Pharmacy, 300041 Timisoara, Romania; cristina.secosan@gmail.com (C.S.); oana.balint@gmail.com (O.B.); aurora1985@yahoo.com (A.I.); ligia_balulescu@yahoo.com (L.B.); grigorasdorin@ymail.com (D.G.); laurentiupirtea@gmail.com (L.P.); 2Center for Modeling Biological Systems and Data Analysis, Department of Functional Sciences, Victor Babeş University of Medicine and Pharmacy, 300041 Timisoara, Romania; dlungeanu@umft.ro; 3Faculty of Medicine, Victor Babeş University of Medicine and Pharmacy, 300041 Timisoara, Romania; 4Department of Anatomy and Embryology, Victor Babeş University of Medicine and Pharmacy, 300041 Timisoara, Romania; dzahoi@umft.ro (D.Z.); amotoc@umft.ro (A.M.)

**Keywords:** colposcopy, HPV, immunohistochemistry, immunostaining, Ki-67, p16

## Abstract

Patients diagnosed with low-grade squamous intraepithelial lesion ((L-SIL) or atypical squamous cells of undetermined significance (ASC-US) are subjected to additional investigations, such as colposcopy and biopsy, to rule out cervical intraepithelial neoplasia 2+ (CIN 2+). Especially in young patients, lesions tend to regress spontaneously, and many human papilloma virus (HPV) infections are transient. Dual-staining p16/Ki-67 has been proposed for the triage of patients with ASC-US or L-SIL, but no prospective study addressing only this subgroup of patients has been conducted so far. We performed a prospective study including all eligible patients referred for a loop electrosurgical excision procedure (LEEP) in the Department of Obstetrics and Gynecology of Timișoara University City Hospital. HPV genotyping and dual-staining for p16/Ki-67 were performed prior to LEEP, at 6 and 12 months after LEEP. A total of 60 patients were included in the study and completed the follow-up evaluation. We analyzed the sensitivity and specificity for biopsy-confirmed CIN2+ using the 95% confidence interval (CI) of high-risk human papilloma virus (HR-HPV), dual-staining p16/Ki-67, colposcopy, and combinations of the tests on all patients and separately for the ASC-US and L-SIL groups. Dual-staining p16/Ki-67 alone or in combination with HR-HPV and/or colposcopy showed a higher specificity that HR-HPV and/or colposcopy for the diagnosis of biopsy confirmed CIN2+ in patients under 30 years. Colposcopy + p16/Ki-67 and HR-HPV + colposcopy + p16/Ki-67 showed the highest specificity in our study.

## 1. Introduction

Cervical cancer is one of the most common gynecological malignancies. Screening programs for cervical cancer have been developed in many countries, most of them being based on cervical cytology (Pap smear). According to the data available in the literature, approximately 30% of patients with cervical cancer may have at least 1 previous false-negative cytological result. Thus, in order to reduce the incidence of the disease, more accurate screening methods are needed [1].

The importance of persistent human papilloma virus (HPV) infection in the etiopathogenesis of cervical cancer draws attention to HPV genotyping as a possible method for primary screening. However, the high rate of transient HPV infections causes HPV genotyping to have low specificity for high-grade cervical lesions, especially in young women [2,3]. Patients diagnosed with low-grade squamous intraepithelial lesion (L-SIL) or atypical squamous cells of undetermined significance (ASC-US) on cytological examination are subjected to additional investigations, such as colposcopy and biopsy, to rule out cervical intraepithelial neoplasia 2+ (CIN2+), according to diagnostic algorithms in many countries. However, many studies show that 70–90% of L-SIL cases regress spontaneously, especially in women under 30 years [4]. This means that many patients undergo unnecessary investigations, such as colposcopy, or even invasive methods, such as biopsy or conization [5].

The need for a better method of triage of these cases, with ASC-US and L-SIL, opens the opportunity for immunohistochemical testing, which may provide additional elements for assessing the severity of the condition. Immunohistochemistry techniques have already been used to overstage cervical cancer samples [6] or to evaluate other gynecologic neoplasms, such as ovarian or endometrial cancer.

Combined immunohistochemical testing of p16/Ki-67 can be used to detect the onset of oncogenesis in cervical cells. P16 overexpression is caused by increased E7 oncoprotein activity (correlated with persistent HPV infection), and Ki-67 is a marker of tumor proliferation. The test is considered positive when both markers are expressed within the same cell. The combined p16/Ki-67 immunohistochemical assay can be performed on the liquid sample medium used for the Pap test and HPV genotyping, and on the biopsies.

According to the latest data from the literature, combined immunohistochemical testing has a comparable sensitivity but a significantly higher specificity compared to HPV testing [7,8,9,10].

More data are needed to validate the use of dual-staining p16/Ki-67 as part of a management algorithm for patients with cytological abnormalities on Pap smear. The prefigured results of the study aim to obtain a new management protocol for patients under the age of 30 years positive for HPV and with cytological abnormalities on the Pap test.

In our study, we update the current evidence regarding the accuracy of p16 staining and dual-staining with p16 and Ki-67 for detecting CIN2+ in the triage of patients with ASC-US or L-SIL under 30 years. Despite many data available up to now regarding the improved specificity of dual-staining in young women with ASC-US or L-SIL compared to HPV genotyping, to our knowledge, this is the first prospective study conducted so far for this specific group of patients. Furthermore, we also compared the accuracy of dual-staining p16/Ki-67 combined with HPV genotyping and colposcopy in order to identity the combination of tests that offers the best accuracy for CIN2+ prediction.

## 2. Materials and Methods

### 2.1. Patient Selection and Inclusion Criteria

We performed a prospective study including all eligible patients that were referred for aloop electrosurgical excision procedure (LEEP) in the Department of Obstetrics and Gynecology of Timișoara University City Hospital, between January 2018 and December 2020. HPV genotyping and dual-staining for p16/Ki-67 were performed prior to LEEP, at 6 and 12 months after LEEP. Inclusion criteria included patients with ASC-US or L-SIL on cervical cytology, patients with indication for LEEP, age under 30 years, no prior cervical procedures (biopsy, LEEP). Exclusion criteria included other cytology results on Pap smear, ongoing pregnancy or postpartum period, and patient age over 30 years. Despite not being an exclusion criterion, none of the patients included in our study had previously been vaccinated for HPV.

### 2.2. Cervical Cytology, Colposcopy and Biopsy

All patients underwent Pap smear testing and colposcopy. Pap smear was performed and evaluated according to the 2001 Bethesda System for reporting cervical cytology. Colposcopy examinations were performed by the same team, with expertise in colposcopy, in all patients, and results were interpreted according to the 2011 International Federation for Cervical Pathology and Colposcopy (IFCPC) terminology system. Only patients with suspected lesions detected by colposcopy were referred for biopsy. LEEP was performed in all cases by the same team of surgeons and biopsy specimens were evaluated by the same pathologist. All patients referred for biopsy were selected for HPV genotyping and dual-staining p16/Ki-67. Both tests were performed prior to LEEP.

### 2.3. HPV Testing

HPV genotyping was performed by the LINEAR ARRAY HPV Genotyping Test [CE-IVD] (ROCHE, Heidelberg, Germany). The HPV test is based on the amplification of target DNA (HPV L1 gene) by multiplex PCR (polymerase chain reaction) and reversed hybridization of the amplified products to a linear array of 37 immobilized probes, which represent different HPV types (6, 11, 16, 18, 26, 31, 33, 35, 39, 40, 42, 45, 51, 52, 53, 54, 55, 56, 58, 59, 61, 62, 64, 66, 67, 68, 69, 70, 71, 72, 73, 81, 82, 83, 84, IS39, and CP6108). PCR was performed on a GeneAmp PCR System 9700 thermal cycler (Applied Biosystems, Waltham, MA, USA), according to the manufacturer’s instructions. Automated hybridization and detection of HPV-DNA were performed on a ProfiBlot 48 Western Blot processor (Tecan Trading AG, Zurich, Switzerland).

### 2.4. Dual-Staining p16/Ki-67 

Immunocytochemistry analysis was performed using the CINtec PLUS Cytology kit (Roche MTM Laboratories, Heidelberg, Germany), according to the manufacturer’s instructions on the cervical Pap smear sample and on the resection piece.

Follow-up visits were scheduled at 6 and 12 months after surgery. Cervical cytology, DNA HPV testing and dual-staining were performed for each patient at each visit.

### 2.5. Statistical Analysis

Apart from the patients’ age, the variables of interest were categorical and were described by the observed frequency (counts) and the corresponding percentage. The Chi-square asymptotic or Fisher’s exact statistical tests were applied to check the statistical significance of the observed differences between groups. The sensitivity and specificity of the diagnostic tests and their combinations were calculated, and the 95% confidence intervals (CIs) were estimated. The analysis was conducted at a 5% level of statistical significance and all reported probability values were 2-tailed.

The data analysis was performed using the statistical software IBM SPSS v. 25 and R v. 4.0.5 packages (including “epiR” v. 2.0.31).

## 3. Results

A total of 60 patients were included in the study and completed the follow-up evaluation. We evaluated the correlation and distribution of p16 and Ki-67 in patients with HPV infection, HR-HPV (high-risk HPV) infection, and negative HPV test and the persistence of HPV, HR-HPV infection, and positive dual-staining at 6 and 12 months after LEEP. We evaluated the correlation between histological grade of low (CIN1), high-grade intraepithelial cervical lesions CIN2+ (CIN2 and CIN3), and in situ carcinoma (CIS) with the immunohistochemical (IHC) expression of p16/Ki-67 and HR-HPV infection as presented in the flow chart (Figure 1).

### 3.1. Demographics

The mean age of the patients was 23.4 years and the age distribution is presented in Figure 2. In total, 37 patients (61.6%) had an L-SIL and 23 patients (38.3%) had an ASC-US result on cervical cytology prior to LEEP.

### 3.2. Colposcopy

Regarding colposcopy, 51 patients (85%) had an abnormal result.

### 3.3. HPV Infection

In total, 52 patients (86.6%) had an HPV infection prior to LEEP, and 36 patients (60%) had an HR-HPV infection. HPV types 16 and 18 were the most frequently encountered, in 22 and 15 patients, respectively.

In total, 21 patients presented a persistent HPV infection at 6 months after LEEP, out of which 20 had HR-HPV. At 12 months after LEEP, 7 patients had a persistent HPV infection, with 6 patients presenting an HR-HPV infection. HPV type 16 was the most frequently encountered in persistent infection, with 7 patients presenting persistent infection at 6 months and 3 patients at 12 months after LEEP (Table 1).

### 3.4. Dual-Staining

Dual-staining for p16/Ki-67 was positive in 31 patients prior to LEEP, in 27 patients on the cervical sample specimen, in 3 patients at 6 months after LEEP, and in 2 patients after 12 months (Table 2).

### 3.5. Histopathology

Regarding the histopathological exam of the cervical specimen, the following results were obtained: 29 patients had CIN I, 18 patients had CIN II, 10 patients had CIN III, and 2 patients had in situ carcinoma.

In total, 36 patients had positive HR-HPV infection: 17 patients CIN I, 8 patients CIN II, 8 patients CIN III, and 2 patients with in situ carcinoma.In total, 31 patients had a positive dual-staining test: 2 patients CIN I, 16 patients CIN II, 10 patients CIN III, and 2 patients had in situ carcinoma.In total, 19 patients had HR-HPV and positive dual-staining: 2 patients CIN I, 7 patients CIN II, 8 patients CIN III, and 2 patients in situ carcinoma.In total, 26 patients had an abnormal colposcopy and HR-HPV: 13 patients CIN I, 7 patients CIN II, and 6 patients CIN III.In total, 24 patients had an abnormal colposcopy and positive dual-staining: 1 patient CIN I, 15 patients CIN II, and 8 patients CIN III.In total, 4 patients had a normal colposcopy and positive dual-staining: 1 patient CIN I, 1 patient CIN II, and 2 patients CIN III.In total, 5 patients had negative HPV and positive dual-staining: 4 patients with CIN II and 1 patient with CIN III.In total, 26 patients had positive HPV and negative dual-staining: 24 patients CIN I and 2 patients CIN II.In total, 16 patients had positive HR-HPV and negative dual-staining: 15 patients CIN I and 1 patient CIN II.

The sensitivity and specificity for biopsy-confirmed CIN2+ was crosstabulated for each test, as shown in Table 3a,b.

For biopsy-confirmed CIN2+, we also analyzed the sensitivity and specificity of HR-HPV, dual-staining p16/Ki-67, colposcopy, and testing combinations on all patients, and separately for the ASC-US and L-SIL groups (Table 4).

For the ASC-US group and L-SIL group and for all the patients, among the tests taken into consideration individually, dual-staining had the best sensitivity and specificity.

Regarding the combination of the three tests for both types of lesion and for the entire group of patients, combination C (colposcopy + p16/Ki-67) revealed the best sensitivity and the best specificity and combination D (HR-HPV + colposcopy + p16/Ki-67).

For the ASC-US group only, combination B (HR-HPV + p16/Ki-67) showed similar results compared with the combinations mentioned before regarding the specificity.

Combination A (HR-HPV + colposcopy) had the lowest sensitivity and specificity of all test combinations. Dual-staining p16/Ki-67, alone or in combination with colposcopy and/or HR-HPV, improved the overall specificity for CIN2+ detection.

Resection margins were negative (in healthy tissue) for all patients and no further surgical treatment was performed during the 12-month follow-up period.

Cervical cytology was normal (NILM) for all patients at the 6- and 12-month follow-up after LEEP. For the patients with persistent HR-HPV infection and/or positive dual-staining at 6 and 12 months, respectively, after LEEP, we recommended a follow-up at a 3-month interval by co-testing (i.e., cervical cytology and HR-HPV detection). No further treatment was applied to date.

## 4. Discussion

### 4.1. Screening for Cervical Cancer by Cervical Cytology and Co-Testing

Screening for cervical cancer is shifting from cervical cytology by Pap smear to HPV genotyping. Data available in the literature suggests that cytology has a sensitivity that ranges between 47% and 62% and specificity between 60% and 95% for the detection of moderate and severe cervical dysplasia and approximately 30% of patients with cervical cancer presented a previous false-negative Pap smear test [1].

The limitations of cervical cytology are certainly linked to subjective interpretation, leading to a wide range of inter-observer variability: approximately, 31% false-positive and 11% false-negative results [11,12]. The highest incidence of inter-observer variability has been noted in patients with ASC-US [13].

On the other hand, HPV genotyping has a higher sensitivity than cytology but a significantly lower specificity, especially in young patients [14].

Since 2004, reports from 6 European randomized control trials (RCTs) have compared HPV and cytology primary screening [15,16,17,18,19,20,21,22,23,24,25]. In a Canadian RCT (CCCaST), HPV and cytology primary testing were performed in combination (co-testing) [26]. In a trial conducted in India, HPV primary screening was compared with cytology primary screening [27]. Four RCTs have published results on the first two screening rounds [16,18,20,23]. The reports suggest that primary screening by co-testing is an effective way to stratify women according to CIN3+ risk. They concluded that co-testing by HR-HPV with cytology triage in women over 30 years is a feasible alternative to screening by cervical cytology.

Co-testing or HPV genotyping has become a standard recommendation in many international guidelines, but it is not recommended for patients under 30 years [28,29,30,31,32].

### 4.2. Age and HPV

The different age distributions has an important impact on the rate of HR-HPV-positive patients. In the ATHENA study, the mean age was 37 years and the overall HR-HPV rate was 32.6%. In contrast, the ASC-US/Low-Grade Squamous Intraepithelial Lesion Triage Study (ALTS study) reported a mean age of 29 years and a rate of 48.0% HR-HPV infection, and in the phase III trial of the Cervista HPV HR test (Hologic), HR-HPV was reported in 57.1% of patients with a mean age of 31 years [33,34]. For the age group 21–29 years, HR-HPV infection was detected in 54.1% of patients with ASC-US compared to 31.3% in the age group 30–39 years and 14.7% in the age group 40–49 years [35]. In our study, the mean age of the patients was 23.4 years.

### 4.3. HR-HPV Testing

The main risk for cervical cancer is represented by persistent infection with HR-HPV types and types 16 and 18 are responsible for the majority of cervical cancer cases [35].

Multiple studies have documented the superior sensitivity of clinically validated HR-HPV testing over Pap cytology for detecting cervical pre-cancer and cancer [14,36,37].

HR-HPV testing for colposcopy triage of patients with L-SIL has been proven to be effective in women ≥30 years but is not recommended in younger women due to high positive rates of HR-HPV infection [38].

Sherman et al. evaluated the sensitivity of HPV testing and repeat cytology for the detection of CIN3 and triage for colposcopy n 3046 patients enrolled in the ALTS study. They concluded that in L-SIL patients, neither HPV genotyping nor repeat cytology is effective for triage. Regarding ASC-US patients, HPV genotyping has proven to be sensitive for CIN3 and cancer detection in older women [34]. Contrary to the high specificity for HPV types 16 and 18, the clinical usefulness is limited due to the contrast between the low prevalence (10.3%) of CIN2+ and higher prevalence of HR-HPV (43%) in ASC-US patients reported in the literature [33,35,39,40].

### 4.4. Studies Regarding the Distribution of HPV Infection

A very high prevalence of HPV infection has been reported in the literature in L-SIL patients (86–97%) and ASC-US (89.5%). HPV infection has also been detected in 27% of patients with NLM results [41]. HR-HPV infection has been reported to be more common in women under 30 years with abnormal cytology, compared to older women [9,42]. Patients with L-SIL are mostly HPV positive, resulting in very poor specificity. Therefore, HPV testing to triage these patients is hardly useful.

In our study, we found the following distribution of HPV infection: 52 patients (86.6%) had HPV infection prior to LEEP, and 36 patients (60%) had HR-HPV infection. In total, 33 out of 37 patients from the L-SIL group had HPV infection prior to LEEP, and 23 had HR-HPV strains. Regarding the ASC-US group, 19 out of 23 patients had HPV infection prior to LEEP, out of which 13 had HR-HPV.

### 4.5. Dual-Staining p16/Ki-67

More specific biomarkers, such as p16 and Ki-67, have been suggested for the triage of patients with L-SIL and ASC-US. Previous publications have demonstrated diffuse staining (over-expression) of p16 to be correlated with the degree of cytological or histological abnormality and represents a hallmark of HPV-dependent transformation [43,44,45,46]. Ki-67 is associated with cell cycle progression and RNA transcription and is over-expressed in CIN2+, adenocarcinoma, and squamous cell carcinoma [47,48] but can also be expressed in benign proliferative lesions [46]. Therefore, a combination of p16/Ki-67 is recommended to increase the specificity for CIN2+ diagnosis, compared to individual assessment of these immunostaining markers [49,50,51,52,53].

Data available in the literature for the triage of ASC-US and L-SIL patients <30 years using dual-staining is insufficient, with most studies being retrospective and do not exclusively address this subgroup of patients. The largest study so far to have reported the results of dual-staining by dividing patients based on age was performed by Bergeron et al. as part of the PALMS study. They reported an increased sensitivity of dual-staining for CIN2+ detection for patients under 30 years and significantly increased positive predictive values for CIN2+ detection compared to HPV genotyping. They also stated that dual-staining triage could considerably reduce the number of colposcopy referrals [33].

### 4.6. Triaging HPV-Positive Women with Normal Cytology by p16/Ki-67 Dual-Staining Cytology Testing

Current reports evaluating the performance of dual-staining p16/Ki-67 in HPV-positive patients (diagnosed by HPV-based screening) with normal cytology (NILM) have revealed that dual-staining detects >70% of underlying CIN3+ and has been proposed for the triage to colposcopy of these patients [54], but it is useful only in older patients, given HPV genotyping is not recommended for screening in patients under 30 years; however, this increases the body of evidence supporting dual-staining as a triage tool for colposcopy.

### 4.7. Impact of Overtreatment

The objectives of cervical cancer screening according to ASCCP 2012 are to prevent morbidity and mortality [2], and to prevent overtreatment of precursor lesions [28].

Several consequences of overtreatment have been identified and need to be taken into consideration: psychological distress in 39% of women, burdening patients with surveillance of an unclear endpoint and efficacy, and risks associated with LEEP (bleeding, infection, adverse obstetrical outcome, e.g., preterm delivery, low birth weight, increased prenatal death due to increased extreme delivery, increase of delivery by C-section) [55,56,57,58].

The ATHENA trial has shown an estimated increase in colposcopies of 40% for patients aged 25–29 years referred to colposcopy based on the ATHENA HPV primary algorithm versus current practice (1 in 5 tested positive for HPV). The ATHENA trial also concluded that substituting cytology with dual-staining triage improved the detection rate of CIN2+ and reduced the number of colposcopies [12]. The PALMS trial reported that referral rates for colposcopy would have been reduced by 50% for patients <30 years with ASC-US if dual-staining was used as a triage tool compared to HPV testing [9].

In our study, the rate of abnormal colposcopies was 85%. The high number of abnormal colposcopies could be explained by the inclusion in our study of only patients with indication for conization.

### 4.8. Persistence of Positive Dual-Staining after Conization

Currently, post-treatment monitoring of patients with CIN is performed by cytology or co-testing (cytology and HR-HPV) [59]. Given the risk of recurrent CIN2 or worse (rCIN2+), dual-staining has been proposed for the surveillance of these patients [60,61].

Polman et al. performed a study on 323 patients treated for CIN in the SIMONATH study in order to evaluate post-treatment monitoring. They performed dual-staining on the residual liquid-based cytology samples obtained at, or shortly after biopsy collection at 6- and/or 12-month follow-up visits. They found that HR-HPV testing had a very high sensitivity and specificity. Regarding co-testing, dual-staining p16/Ki-67 combined with HR-HPV testing showed comparable sensitivity (87.2% vs. 89.7%; ratio 0.97, 95% CI: 0.92–1.03) but significantly higher specificity compared to co-testing by cytology/HR-HPV (74.2% vs. 58.1%; ratio 1.28, 95% CI: 1.19–1.38) [61].

### 4.9. Strengths and Limitations of the Study

The strengths of our study are the prospective design and targeted age group (patients <30 years). Data available in the literature regarding dual-staining for the triage of ASC-US and L-SIL is currently insufficient. The results obtained by our study are promising and could improve the diagnostic algorithm and triage of women <30 years with abnormal cytology of ASC-US/L-SIL.

The main limitation of our study is the rather small number of enrolled patients who completed the follow-up period, rendered by the difficulties of such a prospective design. On the other hand, the design itself brings additional statistical power and reliability of the results. A larger cohort would have certainly generated more precision (i.e., smaller confidence intervals), but the present results are still robust. We are currently continuing the enrollment of patients and will report the data obtained once the follow-up period is complete.

## 5. Conclusions

Dual-staining p16/Ki-67 alone or in combination with HR-HPV and/or colposcopy showed a higher specificity than HR-HPV and/or colposcopy for the diagnosis of biopsy-confirmed CIN2+ in patients under 30 years. Colposcopy+p16/Ki-67 and HR-HPV+ colposcopy+p16/Ki-67 showed the highest specificity in our study. Therefore, we consider that the p16/Ki-67 test could be useful in the triage of young patients with ASC-US or L-SIL and should be taken into consideration for the diagnostic algorithm of this subgroup of patients.

## Figures and Tables

**Figure 1 diagnostics-12-00403-f001:**
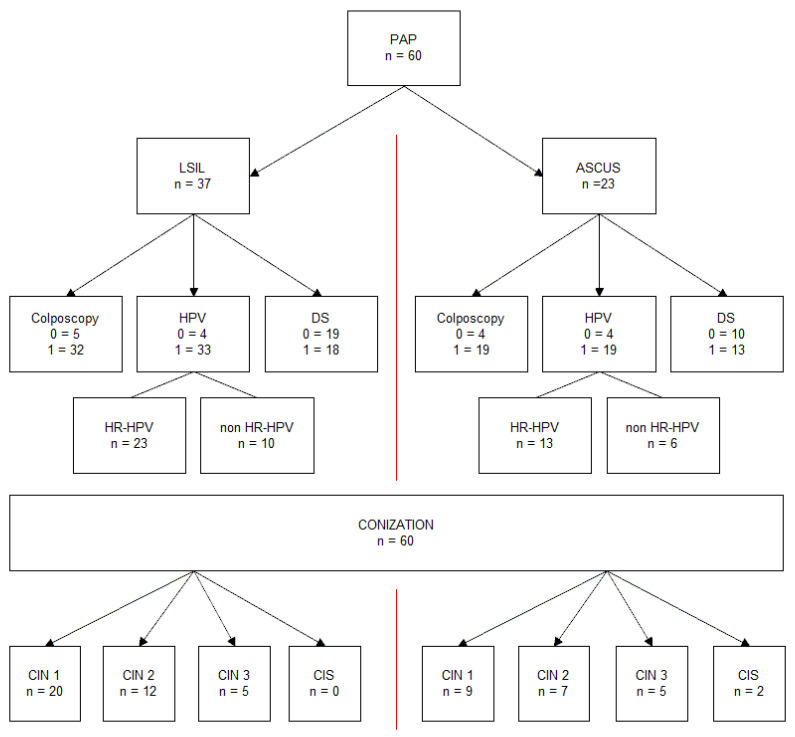
Study flow chart. The patients in the ASC-US and L-SIL groups were followed up for 12 months after LEEP. Abbreviations used: ASC-US: atypical squamous cells of undetermined significance; L-SIL: low-grade squamous intraepithelial lesion; LEEP: loop electrosurgical excision procedure; PAP: cervical cytology, Papanicolaou test; HPV: human papilloma virus; CIN: cervical intraepithelial neoplasia; HR-HPV: high-risk human papilloma virus; CIS: in situ carcinoma; conization: LEEP, loop electrosurgical excision procedure; DS: dual-staining p16/Ki-67.

**Figure 2 diagnostics-12-00403-f002:**
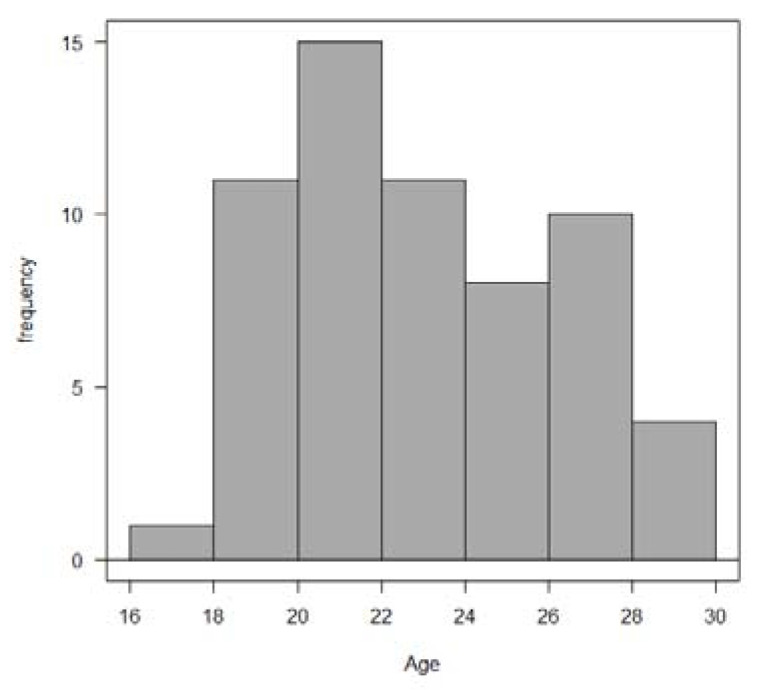
Age distribution of the patients.

**Table 1 diagnostics-12-00403-t001:** HR-HPV infection before conization (bcHR-HPV) at 6 (ac6moHR-HPV) and 12 months after conization (ac12moHR-HPV).

		bcHR-HPV		
Factor	Group	0 (*n* = 24)	1 (*n* = 36)	*p* Value ^(a)^
ac6moHR-HPV ^(a)^	0	24 (100%)	16 (44.4%)	<0.001 **
	1	0	20 (55.6)	
ac12moHR-HPV ^(a)^	0	24 (100%)	30 (83.3%)	0.072
	1	0	6 (16.7%)	

^(a)^ observed frequency (percent); Chi-square asymptotic or Fisher’s exact statistical test. ** statistical significance, *p* < 0.01.

**Table 2 diagnostics-12-00403-t002:** Dual-staining p16/Ki-67 before conization (bcDS) at 6 (ac6moDS) and 12 months after conization (ac12moDS).

		bcDS		
Factor	Group	0 (*n* = 29)	1 (*n* = 31)	*p* Value ^(a)^
ac6moDS ^(a)^	0	26 (89.7%)	28 (90.3%)	1
	1	3 (10.3%)	3 (9.7%)	
ac12moDS ^(a)^	0	29 (100%)	29 (93.5%)	0.492
	1	0	2 (6.5%)	

^(a)^ observed frequency (percent); Chi-square asymptotic or Fisher’s exact statistical test.

**Table 3 diagnostics-12-00403-t003:** Crosstabulation of CIN2+ and Colposcopy, Dual-Staining, HPV and HR-HPV, ASC-US, L-SIL, co-infection, and dual-staining on the conization piece.

**(a). Crosstabulation of CIN2+ and Colposcopy, Dual-Staining p16/Ki-67, HPV, and HR-HPV (Total Number and Separate for Each Strain).**				
		**ConCIN2+**		
**Factor**	**Group**	**0 (*n* = 29)**	**1 (*n* = 31)**	** *p* ** **Value ^(a)^**
Abnormal colposcopy ^(a)^	0	4 (13.8%)	5 (16.1%)	1
	1	25 (86.2%)	26 (83.9%)	
DS ^(a)^	0	27 (93.1%)	2 (6.5%)	<0.001 **
	1	2 (6.9%)	29 (93.5%)	
HPV ^(a)^	0	3 (10.3%)	5 (16.1%)	0.708
	1	26 (89.7%)	26 (83.9%)	
HPV-16 ^(a)^	0	21 (72.4%)	17 (54.8%)	0.188
	1	8 (27.6%)	14 (45.2%)	
HPV-18 ^(a)^	0	21 (72.4%)	24 (77.4%)	0.769
	1	8 (27.6%)	7 (22.6%)	
HPV-31 ^(a)^	0	28 (96.6%)	31 (100%)	0.483
	1	1 (3.4%)	0	
HPV-33 ^(a)^	0	27 (93.1%)	28 (90.3%)	1
	1	2 (6.9%)	3 (9.7%)	
HPV-35 ^(a)^	0	28 (96.6%)	29 (93.5%)	1
	1	1 (3.4%)	2 (6.5%)	
HPV-39 ^(a)^	0	27 (93.1%)	30 (96.8%)	0.606
	1	2 (6.9%)	1 (3.2%)	
HPV-51 ^(a)^	0	25 (86.2%)	30 (96.8%)	0.188
	1	4 (13.8%)	1 (3.2%)	
HPV-53 ^(a)^	0	29 (100%)	30 (96.8%)	1
	1	0	1 (3.2%)	
HPV-56 ^(a)^	0	26 (89.7%)	29 (93.5%)	0.666
	1	3 (10.3%)	2 (6.5%)	
HPV58 ^(a)^	0	28 (96.6%)	31 (100.0%)	0.483
	1	1 (3.4%)	0 (0.0%)	
HPV68 ^(a)^	0	28 (96.6%)	30 (96.8%)	1
	1	1 (3.4%)	1 (3.2%)	
HR-HPV ^(a)^	0	12 (41.4%)	12 (38.7	1
	1	17 (58.6%)	19 (61.3%)	
**(b). Crosstabulation of CIN2+ and ASC-US, L-SIL, co-infection (i.e., more than one strain of HR-HPV present), and dual-staining on the conization piece (conization DS).**				
		**ConCIN2+**		
**Factor**	**Group**	**0 (*n* = 29)**	**1 (*n* = 31)**	** *p* ** **Value ^(a)^**
ASC-US ^(a)^	0	20 (69.0%)	17 (54.8%)	0.298
	1	9 (31.0%)	14 (45.2%)	
L-SIL ^(a)^	0	9 (31.0%)	14 (45.2%)	0.298
	1	20 (69.0%)	17 (54.8%)	
Co-infection ^(a)^	0	17 (58.6%)	18 (58.1%)	1
	1	12 (41.4%)	13 (41.9%)	
Conization DS ^(a)^	0	27 (93.1%)	6 (19.4%)	<0.001 **
	1	2 (6.9%)	25 (80.6%)	

^(a)^ observed frequency (percent); Chi-square asymptotic or Fisher’s exact statistical test. ** statistical significance, *p* < 0.01.

**Table 4 diagnostics-12-00403-t004:** Sensitivity and specificity for biopsy-confirmed CIN2+.

Diagnostic Test	Sensitivity (95% CI)			Specificity (95% CI)		
	All	ASC-US	L-SIL	All	ASC-US	L-SIL
	*n* = 60	*n* = 23	*n* = 37	*n* = 60	*n* = 23	*n* = 37
HR-HPV	0.61 (0.42, 0.78)	0.43 (0.18, 0.71)	0.76 (0.50, 0.93)	0.41 (0.24, 0.61)	0.22 (0.03, 0.60)	0.50 (0.27, 0.73)
Colposcopy	0.52 (0.38, 0.65)	0.79 (0.49, 0.95)	0.88 (0.64, 0.99)	0.14 (0.04, 0.32)	0.11 (0.00, 0.48)	0.15 (0.03, 0.38)
p16/Ki-67	0.94 (0.79, 0.99)	0.93 (0.66, 1.00)	0.94 (0.71, 1.00)	0.93 (0.77, 0.99)	1.00 (0.66, 1.00)	0.90 (0.68, 0.99)
combA	0.45 (0.27, 0.64)	0.21 (0.05, 0.51)	0.65 (0.38, 0.86)	0.55 (0.36, 0.74)	0.33 (0.07, 0.70)	0.65 (0.41, 0.85)
combB	0.58 (0.39, 0.75)	0.43 (0.18, 0.71)	0.71 (0.44, 0.90)	0.93 (0.77, 0.99)	1.00 (0.66, 1.00)	0.90 (0.68, 0.99)
combC	0.77 (0.59, 0.90)	0.71 (0.42, 0.92)	0.82 (0.57, 0.96)	0.97 (0.82, 1.00)	1.00 (0.66, 1.00)	0.95 (0.75, 1.00)
combD	0.42 (0.25, 0.61)	0.21 (0.05, 0.51)	0.59 (0.33, 0.82)	0.97 (0.82, 1.00)	1.00 (0.66, 1.00)	0.95 (0.75, 1.00)

Notation: 95% CI, 95% confidence interval; combA, {HR-HPV + Colposcopy}; combB, {HR-HPV + p16/Ki-67}; combC, {Colposcopy + p16/Ki-67}; combD, {HR-HPV + Colposcopy + p16/Ki-67}.

## Data Availability

Data are available from the first author, upon reasonable request.

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
