# Peer review of "Role of Dual-Staining p16/Ki-67 in the Management of Patients under 30 Years with ASC-US/L-SIL"

_diagnostics, 2022, doi:10.3390/diagnostics12020403_

Round 1

Reviewer 1 Report

MS title:

Role of dual-staining p16/Ki-67 in the management of patients under 30 years with ASC-US/L-SIL

This MS is interesting since it highlights a remarkable potential of dual-staining p16/Ki-67 in the management of patients under 30 years with ASC-US/L-SIL.  

Some modifications of the MS are suggested below [marked in red].

1.In the title - PLEASE, EXPLAIN THIS ABBREVIATION: ASC-US/L-SIL

2.In the Abstract

Patients diagnosed with L-SIL [PLEASE, EXPLAIN THIS ABBREVIATION] or ASC-US [PLEASE, EXPLAIN THIS ABBREVIATION] are subjected to additional investigations, such  as colposcopy and biopsy, to rule out CIN2+ [PLEASE, EXPLAIN THIS ABBREVIATION]. Especially in young patients, lesions tend to regress spontaneously and many HPV infections are transient. Dual-staining p16/Ki-67 has been proposed for the triage of patients with ASC-US or L-SIL, but no prospective study addressing only this subgroup of patients has been conducted so far. We performed a prospective study including all eligible patients referred for LEEP [PLEASE, EXPLAIN THIS ABBREVIATION] in our department PLEASE, PUT THE NAME & LOCATION]. HPV genotyping and dual staining for p16/Ki-67 were performed prior to LEEP at 6 and 12 months after LEEP. A total of 60 patients were included in the study and completed the follow-up evaluation. We analyzed the sensitivity and specificity for biopsy-confirmed CIN2+ using the 95% CI of HR-HPV [PLEASE, EXPLAIN THIS ABBREVIATION], p16/Ki-67 dual-staining, colposcopy, and combinations of tests on all patients and separately for the ASC-US and L-SIL groups. Dual-staining  p16/Ki-67 alone or in combination with HR-HPV and/or colposcopy showed a higher specificity than HR-HPV and/or colposcopy for the diagnosis of biopsy-confirmed CIN2+ in patients under 30 years. Colposcopy+ p16/Ki-67 and HR-HPV + colposcopy + p16/Ki-67 showed the highest specificity in our study.

3.In Figure 1 – Study flow chart. The patients in the ASC-US and L-SIL groups were followed-up for 12 months after LEEP [PLEASE, EXPLAIN ALL THE ABBREVIATIONS  - in the legend].

  1. 3.4. Dual staing [PLEASE, make these CORRECTIONS in many places] = Dual-staining

  1. In the discussion - PLEASE, PUT A SECTION RE: STRENGTHS & LIMITATIONS OF THE STUDY]

  1. In the Conclusions

Dual-staining p16/Ki-67 alone or in combination with HR-HPV and/or colposcopy showed a higher specificity than HR-HPV and/or colposcopy for the diagnosis of biopsy-confirmed  CIN2+ in patients under 30 years. Colposcopy+p16/Ki-67 and HR-HPV+  colposcopy+p16/Ki-67 showed the highest specificity in our study. Therefore, we consider that the p16/Ki-67 test could be useful in the triage of young patients with ASC-US or L-SIL and should be taken into consideration for the diagnostic algorithm of this subgroup of patients. [PLEASE, make these CORRECTIONS in the Conclusions]

  1. PLEASE, PUT AN ABBREVIATION LIST at the end of this MS.

Reviewer 2 Report

Very interesting article

Reviewer 3 Report

This is a prospective study which analyzed the role of dual-staining p16/Ki-67 in the management of patients under 30 years with ASC-US/L-SIL. Currently, there is no sufficient evidence available in literature for the triage of ASC-US and L-SIL patients <30 years by using dual-staining. Thus, the current study demonstrates a good novelty and promising significance for management of women <30 years with abnormal cytology of ASC-US/L-SIL. There are some minor limitations listed below:

1.A total of 60 patients were included in the current study, which is a small number of cohort, thus, it might be more convincing that more patients could be enrolled.

2. A very comprehensive discussion in the Discussion section, however, I would rather not discuss “Non-HPV cervical cancer” and “Dual-staining in pregnancy” which are not correlated to the main subject of this study.
